# Impact of Morphological Characteristics of Green Roofs on Pedestrian Cooling in Subtropical Climates

**DOI:** 10.3390/ijerph16020179

**Published:** 2019-01-09

**Authors:** Gaochuan Zhang, Bao-Jie He, Zongzhou Zhu, Bart Julien Dewancker

**Affiliations:** 1Faculty of Environmental Engineering, The University of Kitakyushu, Kitakyushu 808-0135, Japan; zjhz_zgc@sina.com (G.Z.); bart@kitakyu-u.ac.jp (B.J.D.); 2Faculty of Built Environment, University of New South Wales, Sydney 2052, Australia; 3School of Human Settlements and Civil Engineering, Xi’an Jiaotong University, Xi’an 710049, China; zhuzongzhou@stu.xjtu.edu.cn

**Keywords:** urban heat, pedestrian cooling performance, real neighborhood, extensive green roof, intensive green roof, morphological characteristic, green roof layout, greening coverage ratio, vegetation height, building height

## Abstract

Growing and densifying cities set a challenge for preserving and enhancing green spaces to cool urban spaces. Green roofs, involving the planting of vegetation on rooftops, are regarded as an alternative approach to enhancing urban greenery and urban cooling. For better cooling performances, it is essential to reasonably configure green roofs, especially in real and complex neighborhoods. Therefore, the aim of this paper is to investigate the impact of morphological characteristics of green roofs on pedestrian cooling in real and complex neighborhoods. In specific, based on an ENVI-met model, we studied the effect of greening layout, coverage ratio, vegetation height, and building height on pedestrian air temperature reduction in the tropical city of Hangzhou, China. Results indicate green roofs could generate moderate effects on pedestrian air temperature reduction (around 0.10–0.30 °C), while achieving a cooling performance of 0.82 °C. Green roofs in upwind zones were able to generate the most favorable cooling performance, while green roofs in downwind zones made slight differences to pedestrian thermal environments. Green roofs with a low coverage ratio were not useful for lowering pedestrian temperature, and a greening coverage ratio of 25–75% in upwind zones was cost-effective in real neighborhoods. Locations that were horizontally close to green roofs enjoyed better cooling performances. Increasing vegetation height could strengthen cooling effects of green roofs, while an increase in building height weakened the cooling performance. Nevertheless, higher building height could enhance pedestrian cooling performances because of building shading effects. In addition, because of wind effects and building shading, building height limits for the cooling performance of green roofs could be higher than 60 m.

## 1. Introduction

With ongoing urbanization, worldwide numerous cities, especially megacities, have become congested and overpopulated concrete jungles. Urban problems such as temperature increase, surface flooding, and airborne pollution occur more frequently as a result, thereby influencing the urban ecosystem and living environments [1,2,3,4]. The temperature increase in cities because of the increase in thermal mass and anthropogenic heat, lowered evapotranspiration, and weakened ventilation is a severe and common problem [5,6,7], posing severe challenges in various aspects such as energy consumption, thermal comfort, and citizens’ health conditions [8,9,10]. Nevertheless, the increase in intraurban temperature and thermal stress will be further aggravated as urbanization continues.

Cool material, urban greenery, water bodies, and urban design that can weaken heat source strength and promote excess urban heat dissipation are primary strategies and techniques for urban cooling [11,12,13]. Among them, green roofs, involving the planting of vegetation on building rooftops, is thought as an effective approach due to evapotranspiration and shading effects [14,15,16]. Moreover, green roofs, created through gardens or forests replacing the dark and exposed concrete surfaces, are conducive to energy and carbon reduction through increasing mass and thermal resistance value [17,18,19]. Figure 1 exhibits the cooling mechanism of green roof, where solar heat gain can be reduced by leaves, followed by a conversion of absorbed solar heat to latent heat by evapotranspiration and a reduction of absorbed radiation, resulting in lower surface temperatures and less emitted longwave radiation, and as such, reduced air temperatures [16,18].

To better utilize green roof for urban cooling, various studies have been carried out to examine factors that can influence the cooling potential of green roof [20,21,22,23]. It is shown that the cooling performance varies with climates and geographic conditions. In hot–humid climates (e.g., Hong Kong), the cooling performance underperformed compared with hot–dry climates (e.g., Cairo). Likewise, green roofs in warm-humid climates (e.g., Tokyo) and temperate climates (e.g., Paris) also showed weakened cooling efficiencies [24]. This is because the greenery transpiration is a complex process in relation to various factors within the planetary boundary layer such as solar intensity, wind speed, and soil temperature [24,25]. For instance, in semiarid climates, green roofs could reduce the diurnal sensible heat flux by 150 W/m^2^ and lower the planetary boundary layer height by 700 m [17]. Nevertheless, in temperate continental climate of Chicago, the inclusion of green roofs could reduce surface temperature, but it also reduced wind speed and atmosphere dynamics [18]. Moreover, in subtropical oceanic climate of the Baltimore–Washington metropolitan area, cooling performance of green roof varied with soil moisture, and the cooling performance was negligible when soil moisture was close to its wilting point [19].

Nevertheless, in the same climate, cooling performances of green roof vary with roof structures. Adapting to building types (e.g., single family residential and commercial buildings) with weight, maintenance, and irrigation concerns, green roofs can be divided into extensive green roofs (EGFs) and intensive green roofs (IGFs), as shown in Figure 1. Overall, the IGF is characterized by better cooling performances than the EGF [24]. A study conducted in Hong Kong indicated that IGFs could reduce pedestrian air temperature up to 0.5–1.7 °C, compared with the 0.4–0.7 °C reduced by EGFs [22]. In the same context, another study idealizing the roof height of 20 m also suggested that IGFs had higher cooling efficiencies than EGFs, with pedestrian-level air temperature reductions of 0.6 °C and 0.2 °C, respectively [21]. In addition, various studies on green roof have evidenced that cooling performance could be enhanced with the increase of green coverage ratio [9,21,24]. 

Apart from green roof structures, building characteristics and configurations can also influence cooling effects of green roofs. Overall, the cooling performance decreases along building height, and the cooling effects on pedestrian air temperature are negligible when the building height exceeds 60 m [24]. Meanwhile, the increase of urban density restrains the pedestrian cooling performances, and green roofs play an insignificant role in medium- and high-density neighborhoods [21,24,26]. It is also evidenced that building layout and associated green roof arrangement can affect the cooling performance of green roof [27,28]. For instance, among idealized enclosing-, scattered-, and array-shaped neighborhoods, green roofs with enclosing layout had the best cooling performance, followed by the array layout, and the scattered layout [28]. Along prevailing wind, arranging green roofs in upwind zones could reduce the temperature of the whole neighborhood [28]. 

Overall, above-mentioned studies have suggested that cooling performance of green roofs depends on both building morphology and green roof structures [29]. However, most studies on comparatively investigating cooling performances of green roof are mainly conducted in idealized neighborhoods. In reality, building morphology is quite complex rather than idealized, which affects the microclimate significantly [30,31]. It is essential to further investigate the cooling performance of green roofs in real neighborhoods. Therefore, this paper aims to investigate the impact of morphological characteristics of green roof on pedestrian cooling in real neighborhoods. In specific, this study is conducted in the summer of a subtropical city, Hangzhou (China), for the following objectives (1) to identify relationships between greening layout and corresponding pedestrian cooling performances, (2) to explore the appropriate coverage ratio of green roof in real neighborhoods, (3) to explore the regulation of vegetation height to decrease pedestrian air temperature, and (4) to examine the impact of building height on green roof’s cooling performance in real neighborhoods. 

The remainder of this paper is structured as follows. Section 2 introduces the basic information of the study area and Section 3 describes the field measurement, and settings and calibration of the ENVI-met model. Section 4 analyzes and discusses the impact of greening layout, coverage ratio, vegetation height, and building height on the cooling performance of green roofs, and Section 5 concludes this paper. Overall, this study adds the knowledge of how building morphology and green roof structure influence microclimate simultaneously. The comparative and scientific assessments of green roof cooling performance in real neighborhoods can practically instruct urban planners and policy-makers to choose effective cooling strategies and techniques.

## 2. Study Area

This study was conducted in the context of Hangzhou, the capital city of Zhejiang Province, China (Figure 2). Hangzhou is the center of the Hangzhou metropolitan area in the Yangtze River Delta. The city has a population of more than nine million, covering an area of 16,596 km^2^. 

In the past years, Hangzhou has witnessed a rapid urbanization trend. Its urbanization ratio reached 75.3% by 2016 [32,33]. In recent years, Hangzhou has been undergoing the problem of temperature increase. According to the long-term meteorological data collected at the fixed weather station (Figure 3a, Xiaoshan International Airport, with an elevation of 43 m), its average temperature has increased more than 1 °C in the past 40 years.

Located at 30°15′39″ N and 120°15′26″ E, Hangzhou is characterized by the subtropical climate. Hangzhou has four distinctive seasons every year. Apart from the hot temperature, its summer (June, July and August) is quite humid as the southeast wind can bring a large amount of vapor, and thereby rich precipitation, from the adjacent East China Sea. In particular, the year of 2017 was the hottest year in the past 40 years, as shown in Figure 3a. From Figure 3b, it is observed that July was the hottest month, where the average daily temperature was more than 32 °C. Moreover, the daily minimum and maximum air temperatures were 27 °C and 36 °C, respectively. Overall, the regional climatic conditions make the summer outdoor spaces extremely unfavorable.

## 3. Data and Methodology

This study draws upon field measurements and numerical simulation (based on the computational tool of ENVI-met) to perform the comparative analysis of the impact of green roof morphological characteristics on neighborhood cooling. Figure 4 presents the framework of this study. The field measurements were applied to measure the pedestrian microclimatic conditions of the case study area that was 6 km away from the fixed weather station (Figure 2). These field data collected were utilized to calibrate and validate the numerical model established in the ENVI-met software [34], based on which the microclimates under different scenarios were predicted through changing parameters. 

### 3.1. Field Measurements

Field measurements were carried out around a clothing industrial area consisting of several buildings, as shown in Figure 5. Building heights range between 3 m and 15 m and the average height of buildings is about 10 m. The case study area is seriously insufficient in greenery. There is one willow, four camphor trees and small-scale lawn at the entrance of the factory and the rest land is primarily occupied by traffic land. On the rooftop of these buildings, some scattered pocket green can be observed on cement roofs which have strong reflective and thermal storage capacity.

The study area is used for industrial purpose, so that it is generally operated in the daytime. It is practically meaningful to concentrate on the diurnal microclimates and pedestrian cooling performances of green roofs. Specifically, field measurements were conducted between 8:00 and 20:00 h local time on 29 June 2018. During this period, all buildings were in normal operation. Specifically, we conducted microclimate (including air temperature, relative humidity, wind speed, and wind direction) measurements at five points (in the middle of the road), as presented in Figure 5. The spatial distribution followed the direction of summertime prevailing wind, in order to examine the possible influence of wind on air temperature. All the equipment (as shown in Table 1) was set at the height of 1.4 m above the ground through tripods. Meanwhile, the temperature and relative humidity sensors were covered by aluminum alloy sleeves wrapped in aluminum foil to exclude solar radiation [35], as shown in Figure 5. All the data were recorded every five minutes. Meanwhile, soil temperature was recorded by handheld infrared thermometer (Figure 5).

In addition, we collected the leaf area density (LAD) profiles of the greenery based on leaf area index (LAI)-2000 and hemispherical cameras [36,37]. Specifically, vertical and horizontal profiles of LAD and horizontal canopy structure were measured. The greenery was composed of three-type trees (including Osmanthus trees, sweet viburnum, and glossy privet), which flourished and had a similar height of 3 m. Several kinds of herb grass for which height was less than 1 m were observed on site, as shown in Figure 6. The composition of trees and grasses made a good reference for IGFs and EGFs.

### 3.2. Settings and the Calibration of ENVI-Met Model

ENVI-met software (version 4.3.1) was adopted to estimate the microclimates in the case study area under different scenarios. We chose to simulate a total amount of 30 h from the 00:00 to 06:00 h of the next morning, in which the first six hours were used to achieve microclimate stabilization. In the numerical model, the distribution of grids in the domain model area (270 m × 240 m × 60 m) was 5 m and 2 m in horizontal (Δx and Δy) and vertical (Δz) directions, respectively. Measured buildings were coated by concrete roof and red brick wall with several windows. Five nested grids were adopted to discretize the numerical model. The initial micro-meteorological parameters, simulation controlling parameters, and the definitions of the underlying surface and thermal properties of buildings were set to define the initial boundary conditions, as shown in Table 2.

The meteorological parameters such as wind speed, air temperature and relative humidity, were mainly obtained from field measurements. Model roughness length z0 followed the ENVI-met default values and air moisture content at 2500.0 m was set as 6.5 g/kg. The ENVI-met plant model was divided into 10 equal layers according to vegetation type, so as to facilitate users to define LAD of each layer, thereby accurately describing different canopy shapes and LAD distribution. The setting of plant height in the study area was based on the field survey data, with 1.5 m^2^/m^3^ and 1.0 m^2^/m^3^ for IGFs and EGFs, respectively [38,39]. The parameters of thermal properties of buildings were based on the “Design standard for energy efficiency of residential buildings in hot summer and cold winter zone” in China [40].

In ENVI-met software, the suggested solar radiation ratio (SRR) is 1.0, representing an ideal weather condition with no clouds. However, in realistic environment, cloud that influences the SRR value can be roughly observed. To calibrate the numerical model, therefore, we conducted the sensitivity analysis of various SRR values, including 0.5, 0.6, and 0.7, to the average air temperatures of the five test points in Figure 5. As presented in Figure 7a, the numerical model approached the measured thermal environment much more when the SRR was 0.6. In the scenario of SRR = 0.6, the simulated air temperatures of five test points (8:00–20:00) were compared with the measured air temperatures, as shown in Figure 7b. There was a strong correlation between the measured and simulated air temperatures (R^2^ = 0.8921), meaning the numerical model was in a good agreement with the actual environment [22,24,41,42]. 

### 3.3. Accuracy and Uncertainty of the Numerical Model

Except for SRR, which has been mentioned in Section 3.2, there are some limitations on the setting of surface roughness length and short time scale. The time allowed in ENVI-met model is only between 24 and 48 h, which makes it difficult to obtain the long-term and representative microclimate variation patterns. At the same time, for the wind speed and direction, ENVI-met model adopts less complex input parameters than typical computational fluid dynamics models to mimic real wind fields [43,44]. It was assumed that approaching wind speed and direction remained constant in a whole day in the ENVI-met model. Nevertheless, the approaching wind keeps changing all the time, thereby leading to the deviation of expected air temperatures. Meanwhile, increasing anthropogenic heat emissions from buildings (for example from air-conditioners) discharged into the street canyons can also elevate outdoor temperatures, which is not reflected in the ENVI-met model. At present, moreover, the ENVI-met model cannot define the substrate layer of the green roof, so the thermal effect of soil is ignored. Particularly, the process whereby the wind blows over the soil, taking away near-surface heat, has been neglected [45,46].

### 3.4. Base Model and Data

After the calibration of computational model, we performed various simulations with the variations of greening layout, coverage ratio, vegetation height and building height, based on the weather conditions of 7 July 2017. The air temperature ranged from 28 °C to 36 °C, close to the temperature of the hottest month as described in Section 3.1. Herein we simulated roughly light wind conditions, with the wind speed of 2 m/s and summer prevailing wind direction (southeast) [47,48]. To compare the cooling performance of green roof, we established a base model without any greenery on the rooftop, as shown in Figure 8. Overall, studied the pedestrian thermal environment (1.4 m) at 15:00 h (the hottest time in a day) in different scenarios. For example, Figure 8b presents the pedestrian thermal environment of the base case at 15:00 h. Moreover, we investigated the daily average pedestrian temperatures (07:00 to 06:00 h of the next day) at five test points and variations of average pedestrian temperature of five test points. Based on the base case, the cooling performances of different morphological characteristics of green roof can be derived. To obtain the accurate daily temperature from 07:00 to 06:00 h of the next day, the simulation was run six hours in advance. Due to the limited EGF cooling effects, in Section 4.1 and Section 4.2 only IGFs were adopted.

## 4. Results and Discussion

### 4.1. Effect of Green Roof Layout on Pedestrain Cooling Performance

This section focuses on the cooling performance of five IGF layouts, where the greenery ratio of each scenario is 50%. As shown in Figure 9a, the IGF layouts were divided into five types: Case-Left, Case-Upper, Case-Right, Case-Bottom, and Case-Wind. In the Case-Wind type, the green roof was set in the upwind zones in order to examine the combining effects of prevailing wind and green roof. On the basis of the base case in Section 3.3, the cooling performance of each scenario (at 15:00 h) was obtained and is shown in in Figure 9b. Overall, the maximum cooling performance of IGF could reach up to 0.26 °C. The Case-Wind type witnessed the best spread cooling effects in the local area, while the Case-Left type exhibited the weakest cooling capability to the local area because the green roof was in the downwind zones. Comparatively, the Case-Right and Case-Bottom types had better cooling performances than the Case-Upper type. This indicates the green layout at the upwind side would effectively reduce the air temperature of the entire area. This result is in a good agreement with the existing conclusions that under the wind effects, green roof with orthogonal arrangement can achieve significant cooling effects in downwind area [28,49].

Figure 10 further presents the variations of daily average cooling performance with greening layout. The cooling performance varied dramatically with point location. In particular, the cooling performance were good at point-1, point-2, and point-3, as shown in Figure 10a. Moreover, Case-Wind and Case Bottom types exhibited the best cooling effects at five test points, as shown in Figure 10a. At point-1, the cooling performance of green roof followed the order of Case-Wind > Case-Upper > Case-Left > Case-Bottom > Case-Right. This is because point-1 was in the downwind zones in all scenarios and roof greenery could exert cooling effects on it. At point-2, the green roof in Case-Wind, Case-Bottom, and Case-Right types exhibited their maximum cooling effects, while the green roof in Case-Up and Case-Left types made slight differences because of the upwind location of point-2. Greenery layout exerted the similar impacts on point-3, but the cooling performances were weaker than that on point-2. In addition, the cooling effects of green roof on point-4 and point-5 were quite weak, no more than 0.1 °C. These indicate that both upwind green roof location and its distance from test points (in downwind areas) are important factors determining the pedestrian air temperature reduction. In other words, downwind areas are much cooler and the location closer to green roof is much cooler. 

Moreover, the average air temperature of five test points in all day were calculated to indicate the variations of cooling performances of five scenarios (Figure 10b). Overall, the diurnal fluctuation of cooling performance was more intense than nocturnal one because the great fluctuation of daytime temperature with solar radiation. The temperature reduction resulted from green roof was stable after 17:00 h, indicating green roof could exhibit a long-time cooling performance in the evening. In the daytime, cooling performance decreased rapidly from the morning time to 14:00 h, at which time cooling performance was negligible. This might be because leaf stomata close at high temperatures, and thereby evapotranspiration stops [1,50]. Likewise, Case-Wind was the most prominent scenario in decreasing pedestrian air temperature, followed by Case-Bottom, Case-Right, Case-Left, and Case-Upper. Therefore, strategically installing green roof in upwind zones rather than random greening layouts is more efficient for neighborhood cooling. 

### 4.2. Effect of Coverage Ratio of Green Roof on Pedestrain Cooling Performance 

To further facilitate the application of green roofs, we examined the influence of greening coverage ratio on green roof cooling performances. In particular, five types of IGFs, with the greening coverage ratios of 2%, 25%, 50%, 75%, and 100%, were built, as shown in Figure 11a. Figure 11b presents the cooling performance of each scenario (at 15:00), where cooling performance in Case-2% is negligible and the cooling performance in Case-25% was only about 0.1 °C. With the further increase in greening coverage ratio, the cooling performance was improved and expanded to downward areas simultaneously. For Case-100%, the central area of the neighborhood was cooled up to 0.5 °C. This indicates green roof can form a cool source, like a “cool island”, to isolate the outside heats when the greening area is large enough [22]. In our study, the cooling effects of green roof were not significant as other studies in which cooling performance exceeded 1 °C, which might be due to the larger coverage areas in other studies [51,52,53].

Likewise, the cooling performances of different types of green roof at five test points were examined, as shown in Figure 12. In Figure 12a, a higher greening coverage ratio corresponded to a better cooling performance. Meanwhile, the green roof in all scenarios showed the best cooling performance at the point-1, followed by point-2, point-3, point-4 and point-5. Cooling performance of Case-100% at point-1 exceeded 0.6 °C, while the cooling performances of all green roofs, including Case-100%, at point-5 were less than 0.1 °C consistently. In particular, when the greening coverage ratio was 25%, we can observe that temperature reduction at point-3 was higher than that at other points. This might be because of its shortest horizontal distance from the rooftop greenery (circled zones in Figure 11a).

As shown in Figure 12b, the greening only on one building (Case-2%) was not capable of decreasing surrounding air temperature. With the greening coverage ratio increasing to 25% from 2%, the cooling performance increased. However, the cooling performance of green roof was also limited when greening coverage ratio increased to a certain value. It is indicated that cooling performance of green roof in Case-75% was close to that in Case-100%. There was still an increase in green roof cooling performance with greening coverage ration increasing from 50% to 75%. Therefore, the threshold value of 75% for greening coverage ratio in this study is higher than the suggested value of 50% in Mumbai, India [54], on account of constant greening ratio on the ground. Therefore, in practice, the greenery coverage ratio can be set between 25% and 75% for the aspect of cost-efficiency.

### 4.3. Effect of Vegetation Height of Green Roof on Pedestrain Cooling Performance

Figure 13a presents the physical model and pedestrian cooling performance at 15:00 h of green roofs with different vegetation heights. There were five types of green roof: EGFs with 0.1 m height grass (EGF-0.1M), EGFs with 1 m height grass (EGF-1M), and IGFs with 1 m, 3 m, and 6 m height trees (IGF-1M, IGF-3M and IGF 6M). The coverage ratios of all roofs were 100%. 

From Figure 13b, it is observed that EGF-0.1M, EGF-1M, IGF-1M, IGF-3M, and IGF-6M could yield maximum cooling performances up to 0.10, 0.16, 0.24, 0.53, and 0.61 °C respectively, and the surrounding environment would be visibly improved with the increase in vertical greening height. The increase of greening height can not only provide more shade but also enhance the transpiration by foliage, so that heightening vertical greening is also an effective approach to decreasing the local temperature. 

According to Figure 14a, the cooling performances of green roof at five points showed a constant increase with the increasing vertical greening height. The most significant cooling performance of 0.8 °C could be observed at point-1 when the greening height was 6 m. For the EGFs in this study, however, their cooling performances at point-4 and point-5 were negligible, with a temperature reduction of less than 0.1 °C. Comparing EGF-1M and IGF-1M, the cooling performances of IGF-1M were better than EGF-1M at five points, indicating that different types of green roof generated notably diverse cooling impacts on pedestrian-level temperature [22]. 

Figure 14b presents the variations of average temperature reduction of five test points in a day. Apart from the morning time, all types of green roofs witnessed their best cooling performances at 17:00. This is in the agreement with the phenomenon that a higher leaf area index, a denser and more complex vegetation structure of IGF can weaken more heat strength because of foliage solar shading and passive cooling by evapotranspiration [16,55,56,57]. For this reason, enhancing cooling air production via evapotranspiration and increasing more shade to prevent direct solar exposure are conducive to alleviate local heat stress. Comparing the cooling performance of different types of green roofs, IGFs exhibited better cooling performance than EGFs, while only a slight increase could be observed from IGF-1m to IGF-1M when considering the average temperature reduction of five test points. However, from IGF-1M to IGF-3M and then IGF-6M, a significant increase in cooling performance could be observed. Cooling performance of IGF-6M was about 0.2 °C higher than that of IGF-3M, which was about 0.2 °C higher than that of IGF-1M, as well. Therefore, for better cooling performance, adopting IGFs with higher heights is suggested.

### 4.4. Effect of Building Height on Green Roof Cooling Performance

This section examines the impact of building height on the green roof’s cooling performance. Because of the variation of building height, the dimension of computation domain was altered to 200 m, but the vertical grid (Δz) remained at 2 m. The base case had the average building height of 10 m and covered no vegetation. Afterwards, the same EGFs on three different height building groups were used: E1M-H10 (buildings with average height of 10 m and covered with 1 m grass), E1M-H20 (buildings with average height of 20 m and covered with 1 m grass), and E1M-H30 (buildings with average height of 30 m and covered with 1 m grass). At the same time, same IGFs on four different height building groups were used: I3M-H10 (buildings with average height of 10 m and covered with 3-m trees), I3M-H20 (buildings with average height of 20 m and covered with 3-m trees), I3M-H40 (buildings with average height of 40 m and covered with 3-m trees), and I3M-H60 (buildings with average height of 60 m and covered with 3-m trees), as shown in Figure 15a.

As shown in Figure 15b, cooling performances of green roofs gradually faded away as the increase of building height. This is consistent with the conclusions that as the increase in vertical distance from green roof to the ground, the influence of green roof on pedestrian-level thermal environment decreases [58]. It is worth noting with the increase of building height, EGFs showed larger influencing areas in the downwind zones (e.g., the grey zones in E1M-H10 and E1M-H20), which might be in relation to the shading effects of higher buildings. Same trend could be found in IGF scenarios. Due to stronger evapotranspiration and shading effects of IGFs, the cooling areas were larger than that of EGFs. Nevertheless, this trend will be invalid due to the increase of building height, considering the fact that green roofs exert negligible effects on pedestrian air temperature when building height exceeds 60 m [21]. An existing study has suggested that green roofs are not useful in high-rise area for cooling pedestrian air temperature [21]. However, this conclusion may not be valid, as wind can enhance green roof cooling performance in downwind areas. This has also been evidenced in previous studies via remote sensing technology [51,52].

As shown in Figure 16a, the cooling effects of green roof decreased with the increase of building height. At point-1, the cooling effects were gradually weakened by increasing building height, and there were slight differences among the cooling performances of I3M-H10, I3M-H20, and I3M-H40. However, at point-2 and point-3, I3M-H10, I3M-H20, and I3M-H40 showed an increasing cooling performance difference. At other points, the cooling performances of green roofs were negligible. Moreover, cooling performances of different green roofs decreased gradually from point-1 to point-5. In particular, the cooling performance of I3M-H10 at point-1 reached 0.6 °C, while it decreased to negligible (less than 0.1 °C) at point-5. For I3M-H60, it could exert cooling effects on point-1 and point-2, with the values of 0.4 °C and 0.2 °C, respectively, while at point-3, point-4, and point-5, the cooling performances were negligible. 

Figure 16b also shows green roofs with higher building height had weaker cooling effects on pedestrian air temperature. In the night time, I3M-H10 exhibited stable cooling effects around 0.35 °C, about 0.2 °C higher than that of I3M-H-60. Likewise, E1M-H10 generated the cooling performance of 0.2 °C, followed by that of E1M-H20 (about 0.1 °C), and E1M-H40 (about 0.05 °C). At the time of sunrise and sunset, green roofs achieved their peak cooling performance. However, at 14:00 h, lower buildings (E1M-H10 and I3M-H10) exhibited lower cooling performances, as compared with other EGFs and IGFs built on higher buildings. This may be because more solar radiation can enter the bottom of shallow street canyon, while higher buildings can shelter the ground from solar radiation [7]. Therefore, in the daytime, the pedestrian cooling performance of green roofs is affected by both building shading effects and vegetation shading and evapotranspiration effects. 

In addition, in the scenario of I3M-H60, it is of interest to observe that cooling performances of green roof at point-1 and point-2 could reach 0.42 °C and 0.24 °C, respectively (Figure 16a). This result is in contrast with the conclusion that green roofs play a negligible role in pedestrian temperature reduction when building height exceeds 60 m [24]. Likewise, at point-1 and point-2, the cooling performance of green roof could reach 0.2 °C and 0.1 °C in the scenario of E1M-H40 (Figure 16a). It is concluded that the height limit of buildings upon which green roof is located varies with target locations, which is in relation to the wind effects [28,49]. Based on the average temperature of five points at 14:00 h (Figure 16b), we can further conclude that the building orientation, through influencing the pedestrian-level solar exposure, becomes another factor affecting the building height limit, exceeding which green roofs show no differences in terms of cooling performance [7].

### 4.5. Cooling Performances in All Scenarios

Above sections have analyzed the impacts of greening layout, coverage ratio, vegetation height, and building height on pedestrian cooling performances of green roofs. Table 3 further summarizes the air temperature reduction at five points under different scenarios. From the temperature reduction at five points, it is observed overall that green roofs exerted moderate cooling effects on the environment at the pedestrian level compared with other cooling strategies and techniques of cool pavements, water bodies, and urban forests. Nevertheless, green roofs could also generate the most favorable cooling performance of 0.82 °C at the point-1 in the IGF-6M scenario. Green roofs could generally generate cooling performances of 0.10–0.30 °C at point-1, point-2, and point-3, while the cooling performances were primarily less than 0.10 °C at point-4 and point-5. Meanwhile, temperature reduction generally followed the pattern of point-1 > point-2 > point-3 > point-4 and point-5, except for Case-Right, Case-Bottom, Case-Wind, and Case-25%. This further shows that downwind side is generally the most advantageous part to enjoy the cooling performance of green roofs [28,49]. From the average temperature of five points, it is found that the cooling performances (*T_24h_*) were generally low, at around 0.10–0.30 °C. Nevertheless, Case-Wind, Case-100%, IGF-6M, and I3M-H10 were the most favorable scenarios in different categories, with *T_24h_* of 0.26, 0.35, 0.46, and 0.35 °C, respectively. For practically enhancing pedestrian cooling performance of green roofs, therefore, it should combine the conditions of wind, higher coverage, higher vegetation height, and lower building height. 

## 5. Conclusions

This paper investigated the effect of morphological characteristics (greening layout, greening coverage ratio, vegetation height, and building height) of green roofs on pedestrian cooling performances in a real neighborhood in the subtropical city of Hangzhou, based on field measurement and numerical simulations. Based on the analysis in this study, the following conclusions can be drawn.
Overall, green roofs could generate a moderate cooling performance at the pedestrian level, while the most favorable cooling performance could reach up to 0.82 °C. To better utilize green roofs for pedestrian cooling, it is essential to simultaneously control the wind, greening layout, coverage ratio, vegetation height, and building height.Installing green roofs in upwind zones was favorable for pedestrian-level cooling, while green roofs in the downwind zones could only exert limited cooling effects. Overall, the cooling performance of green roof followed the pattern of Case-Wind > Case-Upper > Case-Left > Case-Bottom > Case-Right.A green roof with a low greening coverage ratio was not useful to improve pedestrian thermal environment. The cooling performance increased with the increasing coverage ratio, but the cooling performance reached a threshold when the coverage ratio increased to a certain value. Nevertheless, a neighborhood with a high coverage ratio could experience a “cool island” in the central area. In addition, the horizontal distance from green roofs to the target location could also influence the pedestrian cooling performance, where a short distance corresponded to a better cooling performance.Vegetation height played a critical role in improving green roof cooling performance. IGFs exhibited better cooling performances than EGFs, and the increase in vegetation height resulted in better cooling performances. The cooling effects of IGF-6M on the whole area could reach 0.5 °C, and more than 0.3 °C at 14:00 h. However, when greening height was under 1 m, the cooling effects of green roofs were insignificant. Building height was also an important factor affecting green roof cooling performance. With the increase of building height, the cooling effects of green roofs generally showed a trend of decrease. At this time, however, buildings and vegetation had combined effects, where higher buildings and vegetation could generate stronger cooling effects at the noon time. Moreover, because of wind effects and building shading, the building height limit for the cooling performance of green roofs was increased.


## Figures and Tables

**Figure 1 ijerph-16-00179-f001:**
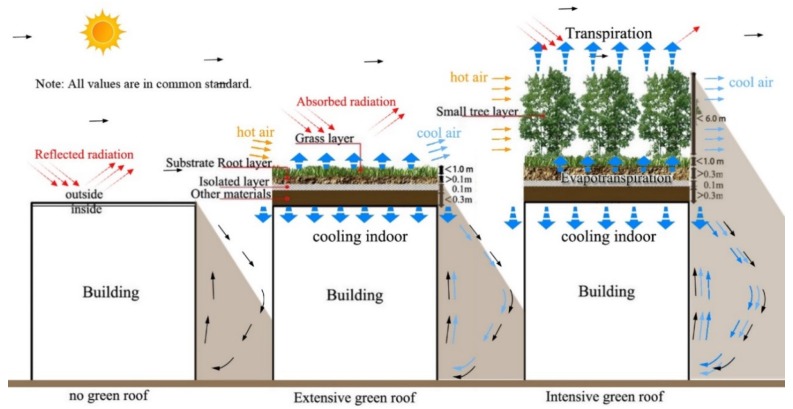
A schematic structure of an extensive green roof and an intensive green roof, and the cooling mechanism in the daytime.

**Figure 2 ijerph-16-00179-f002:**
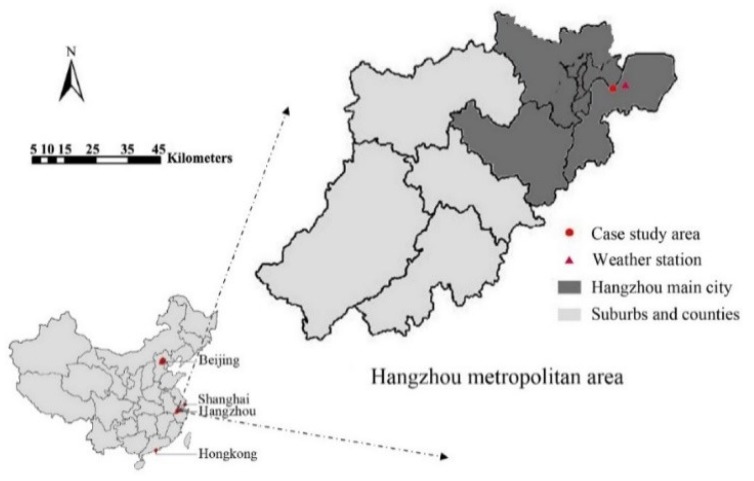
Location of Hangzhou city and the fixed weather station.

**Figure 3 ijerph-16-00179-f003:**
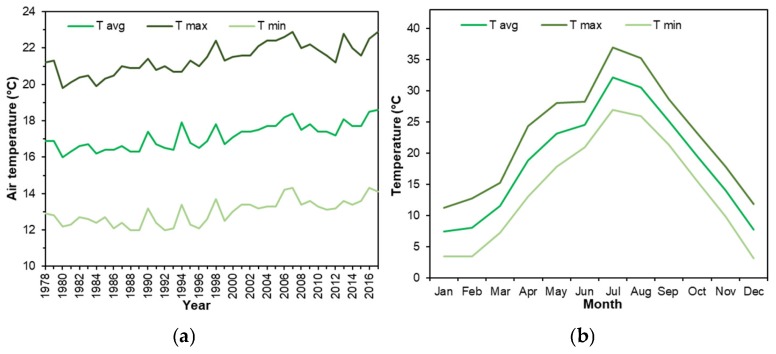
(**a**) Meteorological condition of Hangzhou city (1978–2017). (**b**) Weather condition of Hangzhou in the year of 2017. (Note: T avg: average air temperature, T max: average maximum air temperature, T min: average minimum air temperature).

**Figure 4 ijerph-16-00179-f004:**
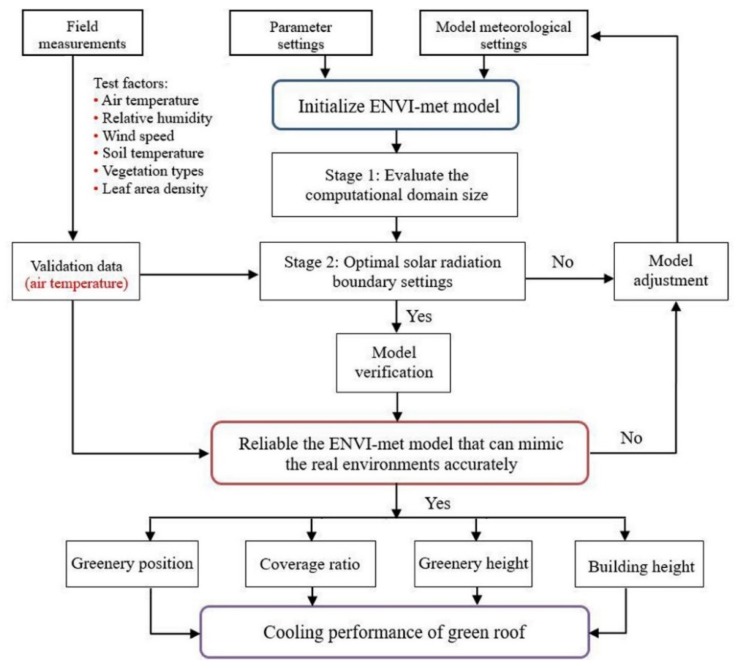
A framework for comparative analyzing cooling performance of green roofs under different scenarios.

**Figure 5 ijerph-16-00179-f005:**
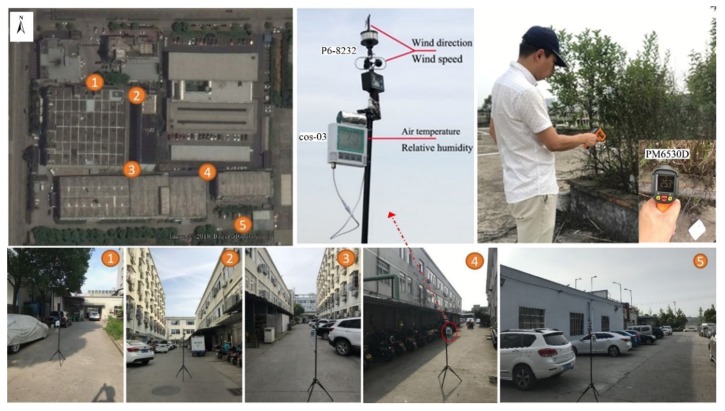
On-site field measurement and the spatial distribution of test points.

**Figure 6 ijerph-16-00179-f006:**
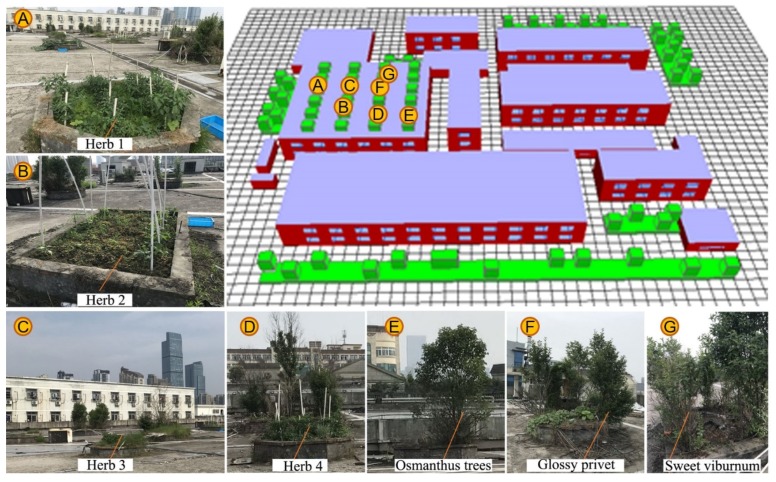
Photos of rooftop vegetation and the initial ENVI-met model.

**Figure 7 ijerph-16-00179-f007:**
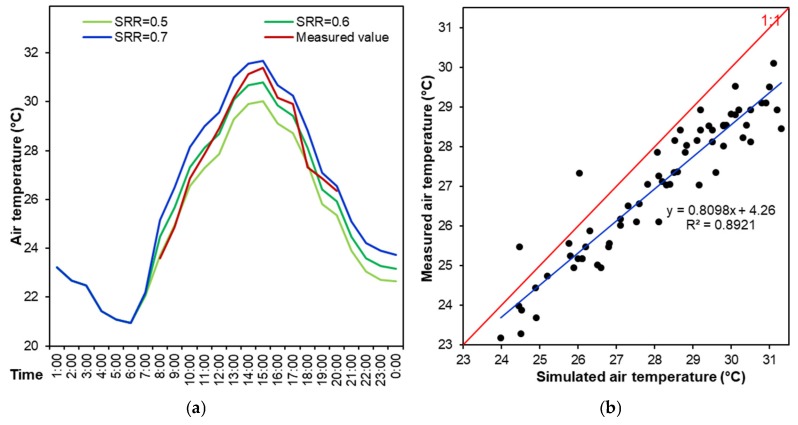
Calibration of ENVI-met model. (**a**) Comparison of measured average air temperature of five points with numerical results in different scenarios of solar radiation, and (**b**) Correlations between measured and simulated air temperatures of five points (from 08:00 to 20:00 h).

**Figure 8 ijerph-16-00179-f008:**
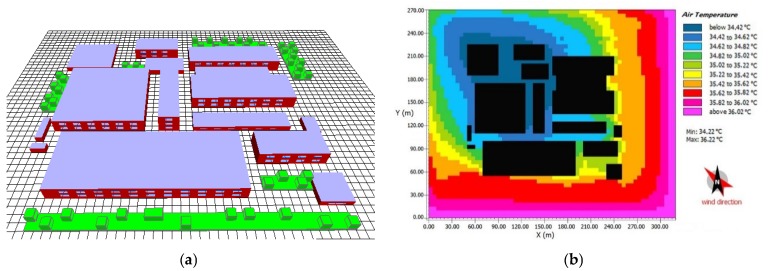
Base case diagram in ENVI-met software: (**a**) Physical model and (**b**) pedestrian thermal environment at 15:00 h.

**Figure 9 ijerph-16-00179-f009:**
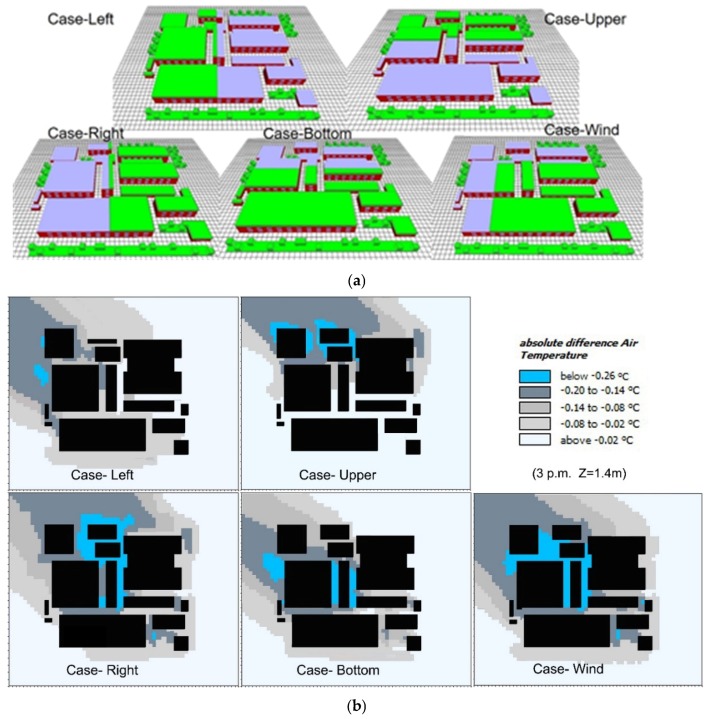
Comparison of the impact of greening layout on green roof cooling performance (**a**) Physical models, and (**b**) pedestrian air temperature reduction at 15:00 h.

**Figure 10 ijerph-16-00179-f010:**
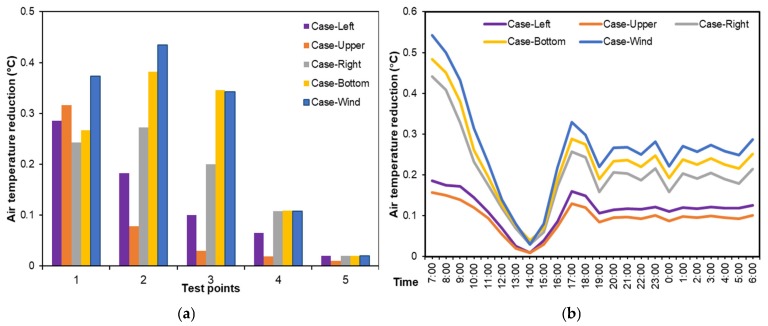
Average air temperature reduction in different green roof layout scenarios. (**a**) Daily average air temperatures at five test points, and (**b**) daily variations of average air temperature of five test points from 07:00 to 06:00 h of the next day.

**Figure 11 ijerph-16-00179-f011:**
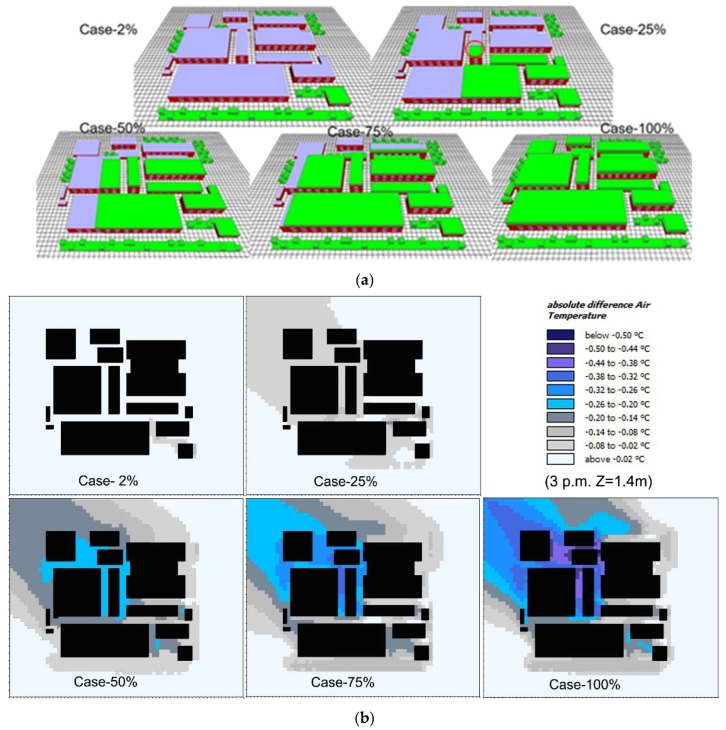
Comparison of the impact of greening coverage ratio on green roof cooling performance (**a**) Physical models, and (**b**) pedestrian air temperature reduction at 15:00 h.

**Figure 12 ijerph-16-00179-f012:**
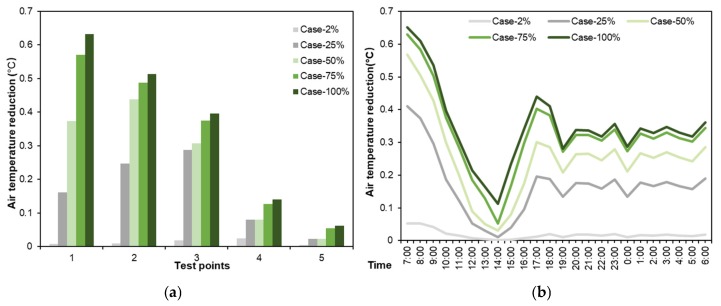
Average air temperature reduction in different greening coverage ratio scenarios. (**a**) Daily average air temperatures at five test points, and (**b**) daily variations of average air temperature of five test points from 07:00 to 06:00 h of the next day.

**Figure 13 ijerph-16-00179-f013:**
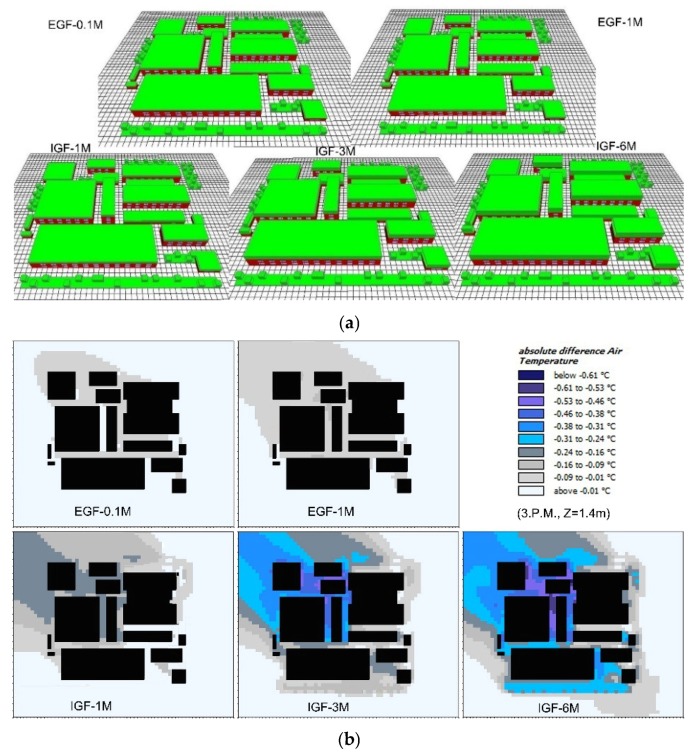
Comparison of the impact of vertical greening height on green roof cooling performance (**a**) Physical models, and (**b**) pedestrian air temperature reduction at 15:00 h.

**Figure 14 ijerph-16-00179-f014:**
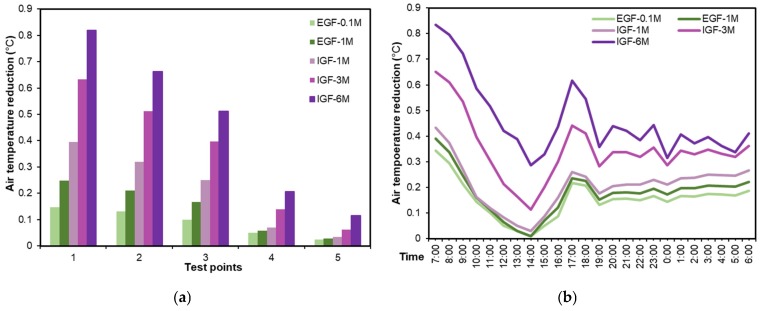
Average air temperature reduction in different vegetation height scenarios. (**a**) Daily average air temperatures at five test points, and (**b**) daily variations of average air temperature of five test points from 07:00 to 06:00 hof the next day.

**Figure 15 ijerph-16-00179-f015:**
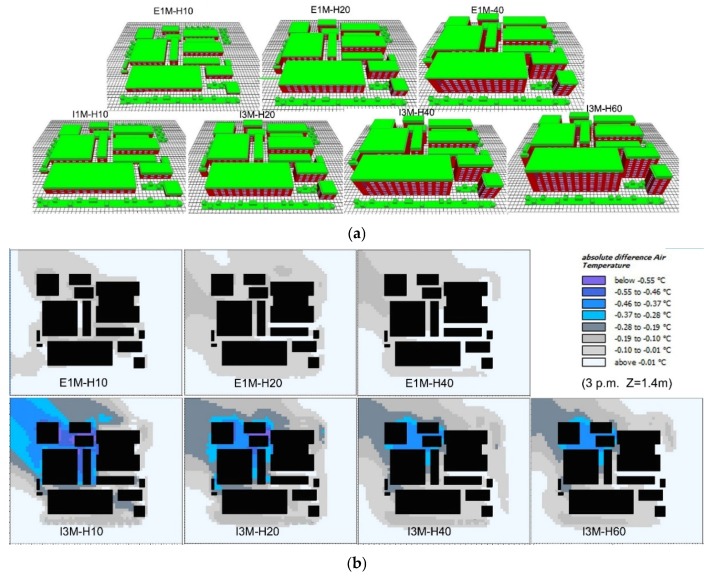
Comparison of the impact of building height on green roof cooling performance (**a**) Physical models, and (**b**) pedestrian air temperature reduction at 15:00 h.

**Figure 16 ijerph-16-00179-f016:**
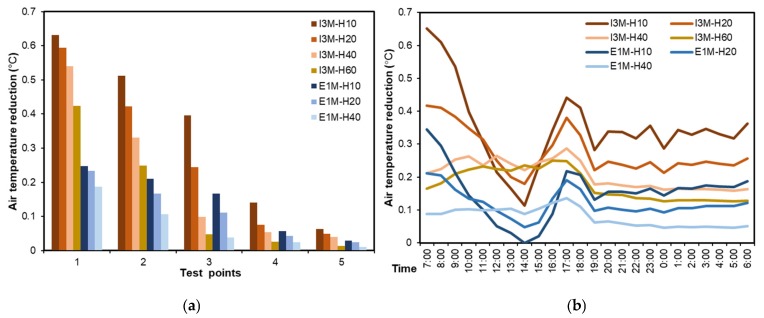
Average air temperature reduction in different building height scenarios. (**a**) Daily average air temperatures at five test points, and (**b**) daily variations of average air temperature of five test points from 07:00 to 06:00 h of the next day.

**Table 1 ijerph-16-00179-t001:** Parameters and instruments used during the field measurements. LAI: leaf area index.

Item	Instrument	Parameter	Resolution/Range	Frequency
Temperature	Cos-03	Air temperature	±0.1 °C (−20 °C–60 °C)	5 min
Humidity	Cos-03	Relative humidity	±1.5% (0–100%)	5 min
Wind	P6-8232	Wind speed Wind direction	±0.9 m/s (0–30 m/s) ±0.5° (0–360°)	30 min
Soil	PM 6530D	Soil temperature	±0.5 °C (−20 °C–60 °C)	1 h
LAI	LAI-2000	Leaf area density	2.5 m CEP (50% deviation)	1 D

CEP: Circular Error Probable; LAI: leaf area index.

**Table 2 ijerph-16-00179-t002:** Settings of boundary condition in verified ENVI-met model.

Item	Parameter	Value
Meteorological	Solar radiation	0.5; 0.6; 0.8
parameters	Initial wind direction	45°(SE)
	Wind speed at 10 m	2 m/s
	Initial air temperature	22.0 °C
	Relative humidity	71%
	Air moisture content (2500.0 m)	6.5 g·kg^−1^
	Roughness length	0.1 m
Roof	LAD of IGFs LAD of EGFs	1.5 m^2^·m^−3^ 1.0 m^2^·m^−3^
	Average albedo of green/roof	0.2/0.3
Soil	Initial surface temperature/humidity (0–20 cm)	25.0 °C/50%
	Initial temperature/humidity in middle depth (0–20 cm) Initial temperature/humidity in deep depth (>50 cm)	26.0 °C/60% 26.0 °C/60%

LAD: leaf area density; IGFs: intensive green roofs; EGFs: extensive green roofs.

**Table 3 ijerph-16-00179-t003:** A summary of pedestrian cooling performance of green roof in different scenarios.

Scenarios		Daily Average Temperature (°C)					Average Temperature of Five Points (°C)		
		1	2	3	4	5	*T_24h_*	*T_max_*	*T_min_*
Layout	Left	0.29	0.18	0.10	0.07	0.02	0.13	0.29	0.02
	Upper	0.32	0.08	0.03	0.02	0.01	0.09	0.32	0.01
	Right	0.24	0.27	0.20	0.11	0.02	0.17	0.27	0.02
	Bottom	0.27	0.38	0.35	0.11	0.02	0.22	0.38	0.02
	Wind	0.37	0.43	0.34	0.11	0.02	0.26	0.43	0.02
Coverage ratio	Case 2%	0.01	0.01	0.02	0.02	0.01	0.01	0.02	0.01
	Case 25%	0.16	0.25	0.29	0.08	0.02	0.16	0.29	0.02
	Case 50%	0.37	0.44	0.31	0.08	0.02	0.24	0.44	0.02
	Case 75%	0.57	0.49	0.38	0.13	0.05	0.32	0.57	0.05
	Case 100%	0.63	0.51	0.40	0.14	0.06	0.35	0.63	0.06
Vegetation height	EGF-0.1M	0.15	0.13	0.10	0.05	0.03	0.10	0.15	0.03
	EGF-1M	0.25	0.21	0.17	0.06	0.03	0.15	0.25	0.03
	IGF-1M	0.39	0.32	0.25	0.07	0.04	0.21	0.39	0.04
	IGF-3M	0.63	0.51	0.40	0.14	0.06	0.35	0.63	0.06
	IGF-6M	0.82	0.66	0.51	0.21	0.12	0.46	0.82	0.12
Building height	E1M-H10	0.25	0.21	0.17	0.06	0.03	0.15	0.25	0.03
	E1M-H20	0.23	0.17	0.11	0.04	0.02	0.12	0.23	0.02
	E1M-H40	0.19	0.11	0.04	0.02	0.01	0.07	0.19	0.01
	I3M-H10	0.63	0.51	0.40	0.14	0.06	0.35	0.63	0.06
	I3M-H20	0.59	0.42	0.24	0.08	0.05	0.28	0.59	0.05
	I3M-H40	0.54	0.33	0.10	0.05	0.04	0.21	0.54	0.04
	I3M-H60	0.42	0.25	0.05	0.03	0.01	0.15	0.42	0.01

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
