# Peer review of "Impact of Morphological Characteristics of Green Roofs on Pedestrian Cooling in Subtropical Climates"

_ijerph, 2019, doi:10.3390/ijerph16020179_

Round 1

Reviewer 1 Report

Nice paper, starts well, is easy to read and is an interesting piece of research; which is both timely and topical, however it is misleading in the sense that whilst the results clearly demonstrate that green roofs have an impact on pedestrian air temperatures here these are minimal bordering on negligible and short lived and this is not made clear.  The limitations of ENVI-met need to be introduced much earlier in the text, and this needs to include the short time scale the tool covers.. i.e. a 24 hour period. The paper reports on the cities UHI but this is irrelevant to the paper which reports on the intraurban microscale daytime air temperature effects whereas the UHI is a citywide (often) nocturnal phenomenon, whish whilst made up of a series of intraurban microscale effects is not helpful here. I would recommend that all UHI comments are removed and just the intraurban air temperature effects are discussed, and that these are discussed in terms of the timing of the timing of the building groups function i.e., commercial/office group with a daytime function OR residential group with mostly a nocturnal function and how these effects might impact on night-time overheating risks, and if this is relevant to the study. Overall, I like the paper… however the impacts of vertical and horizontal distance greening from target location needs to be better explained against the site specific background climate conditions.

Author Response

To reviewer #1,

Nice paper, starts well, is easy to read and is an interesting piece of research; which is both timely and topical, however it is misleading in the sense that whilst the results clearly demonstrate that green roofs have an impact on pedestrian air temperatures here.

Response: Many thanks go to your valuable and we have revised the paper according to them one by one.

Q1: these are minimal bordering on negligible and short lived and this is not made clear. 

Response 1: This problem has been addressed. We have added the overall assessments of the pedestrian cooling performances of green roof in the abstract (Line 20-22), discussion (Line 429-435, Line 439-442) and conclusion (Line 451-456).

Q2: The limitations of ENVI-met need to be introduced much earlier in the text, and this needs to include the short time scale the tool covers.. i.e. a 24 hour period.

Response 2: This problem has been addressed. We have moved to Section 3.3. Meanwhile, we have added the limitation of time scale in the line of 218-220.

Q3: The paper reports on the cities UHI but this is irrelevant to the paper which reports on the intraurban microscale daytime air temperature effects whereas the UHI is a citywide (often) nocturnal phenomenon, whish whilst made up of a series of intraurban microscale effects is not helpful here. I would recommend that all UHI comments are removed and just the intraurban air temperature effects are discussed.

Response 3: This problem has been addressed. The UHI effects have been revised as urban cooling.

Q4: and that these are discussed in terms of the timing of the timing of the building groups function i.e., commercial/office group with a daytime function OR residential group with mostly a nocturnal function and how these effects might impact on night-time overheating risks, and if this is relevant to the study.

Response 4: This problem has been addressed, where we have mentioned the function of buildings (Line 152-155).

Q5: Overall, I like the paper… however the impacts of vertical and horizontal distance greening from target location needs to be better explained against the site specific background climate conditions.

Response 5: This problem has been addressed. We have further discussed the horizontal distance greening from target location (Line 312-315).  Meanwhile, the vertical distance greening from target location has been mentioned (Line 413-422).

Reviewer 2 Report

Manuscript ID ijerph-409983 

Impact of Morphological Characteristics of Green Roof on Pedestrian Cooling in Subtropical Climates; By Gaochuan Zhang , Bao-Jie He * , Zongzhou Zhu , Bart Julien Dewancker

OVERALL COMMENT:

The authors have attempted to study the impact of green roofs on pedestrian cooling in subtropical climate using ENVI-met model over a small domain in Hangzhov, China. Authors test the sensitivity of different green roof locations and percentage coverage within their domain. I overall liked the design of experiments, but I am concerned with the presentation of the article. I found multiple English language mistakes related to grammar and sentence structures.  I read the Introduction section and provide comments and guidance to authors to improve the writing further. 

Recommendation: Major Revisions

Detailed comments:

I provide a pdf of my hand-written comments on the article that will guide and help authors improve the manuscript. The abstract is written poorly with poor and unclear sentences. 

Introduction: I would recommend discussing the impact pathways of green roofs within the boundary layer. You try to mention the impact while in passing on Lines 62-63. Studies have shown that the green roofs can substantially modify the urban boundary layer. Here are the studies I recommend authors to review for different climates. 

1.    Song, J., Wang, Z.H. and Wang, C., 2018. The Regional Impact of Urban Heat Mitigation Strategies on Planetary BoundaryLayer Dynamics over a Semiarid City. Journal of Geophysical Research: Atmospheres.

2.    Sharma, A., Conry, P., Fernando, H.J.S., Hamlet, A.F., Hellmann, J.J. and Chen, F., 2016. Green and cool roofs to mitigate urban heat island effects in the Chicago metropolitan area: Evaluation with a regional climate model. Environmental Research Letters, 11(6), p.064004.

3.    Li, D., Bou-Zeid, E. and Oppenheimer, M., 2014. The effectiveness of cool and green roofs as urban heat island mitigation strategies. Environmental Research Letters, 9(5), p.055002

Lines 88-90: Not all ENVI-met studies are performed for idealized conditions. For example, a study in Chicago used realistic morphology and large-scale forcing to evaluate the impact of building characteristics on local neighborhood flows, pedestrian comforts, and their energy demand in current and future climate. 

Conry, P., Sharma, A., Potosnak, M. J., Leo, L. S., Bensman, E., Hellmann, J. J., & Fernando, H. J. (2015). Chicago’s heat island and climate change: Bridging the scales via dynamical downscaling. Journal of Applied Meteorology and Climatology, 54(7), 1430-1448.

You missed citing the ENVI-met developer's article, Bruse and Fleer 1998.  

Bruse, M. and Fleer, H., 1998. Simulating surface–plant–air interactions inside urban environments with a three dimensional numerical model. Environmental modelling & software13(3-4), pp.373-384.

Figure 1: Caption incomplete. 

Overall, there has been a recent review article on different roofing methods on roof designs and climate responsiveness that will provide a good reference for you. 

Abuseif, M. and Gou, Z., 2018. A Review of Roofing Methods: Construction Features, Heat Reduction, Payback Period and Climatic Responsiveness. Energies, 11(11), p.3196.

Good luck!

Author Response

To reviewer 2#

OVERALL COMMENT:

Q1: The authors have attempted to study the impact of green roofs on pedestrian cooling in subtropical climate using ENVI-met model over a small domain in Hangzhov, China. Authors test the sensitivity of different green roof locations and percentage coverage within their domain. I overall liked the design of experiments, but I am concerned with the presentation of the article. I found multiple English language mistakes related to grammar and sentence structures.  I read the Introduction section and provide comments and guidance to authors to improve the writing further.

Recommendation: Major Revisions

Response 1: We have revised this paper and further polished the English.

Q2: Detailed comments:

I provide a pdf of my hand-written comments on the article that will guide and help authors improve the manuscript. The abstract is written poorly with poor and unclear sentences.

Response 2: Many thanks go to your comments, we have revised them according to your guidance, and the references you mentioned have been added. Please see the references 17, 18, 19 and 23.

Q3: Introduction: I would recommend discussing the impact pathways of green roofs within the boundary layer. You try to mention the impact while in passing on Lines 62-63. Studies have shown that the green roofs can substantially modify the urban boundary layer. Here are the studies I recommend authors to review for different climates.

1.    Song, J., Wang, Z.H. and Wang, C., 2018. The Regional Impact of Urban Heat Mitigation Strategies on Planetary BoundaryLayer Dynamics over a Semiarid City. Journal of Geophysical Research: Atmospheres.

2.    Sharma, A., Conry, P., Fernando, H.J.S., Hamlet, A.F., Hellmann, J.J. and Chen, F., 2016. Green and cool roofs to mitigate urban heat island effects in the Chicago metropolitan area: Evaluation with a regional climate model. Environmental Research Letters, 11(6), p.064004.

3.    Li, D., Bou-Zeid, E. and Oppenheimer, M., 2014. The effectiveness of cool and green roofs as urban heat island mitigation strategies. Environmental Research Letters, 9(5), p.055002

Response 3: We have analyzed the impact of green roof in urban boundary layer. Please be kind to check Line 64-72. References you mentioned have been added to support our paper.

Q4: Lines 88-90: Not all ENVI-met studies are performed for idealized conditions. For example, a study in Chicago used realistic morphology and large-scale forcing to evaluate the impact of building characteristics on local neighborhood flows, pedestrian comforts, and their energy demand in current and future climate.

Conry, P., Sharma, A., Potosnak, M. J., Leo, L. S., Bensman, E., Hellmann, J. J., & Fernando, H. J. (2015). Chicago’s heat island and climate change: Bridging the scales via dynamical downscaling. Journal of Applied Meteorology and Climatology, 54(7), 1430-1448.

Response 4: We have adjusted our description. Please be kind to check Line 83-97.

Q5: You missed citing the ENVI-met developer's article, Bruse and Fleer 1998. 

Bruse, M. and Fleer, H., 1998. Simulating surface–plant–air interactions inside urban environments with a three dimensional numerical model. Environmental modelling & software, 13(3-4), pp.373-384.

Response 5: It has been added, please check reference 34.

Q6: Figure 1: Caption incomplete.

Response 6: This has been revised, Line 57-58.

Q7: Overall, there has been a recent review article on different roofing methods on roof designs and climate responsiveness that will provide a good reference for you.

Abuseif, M. and Gou, Z., 2018. A Review of Roofing Methods: Construction Features, Heat Reduction, Payback Period and Climatic Responsiveness. Energies, 11(11), p.3196.

Response 7: We have read this paper and added it in the paper Reference 23.  

Reviewer 3 Report

This paper aims to understand the impact of green roofs on the outdoor thermal comfort. The paper is really interesting, leading to an important topic for the sustainable urban development. The study is quite comprehensive, combining both the measurements with the simulations with ENVImet.

In order to evaluate the impact of green roofs on the outdoor thermal comfort, the air temperature is computed. As you have already the model in ENVImet, can you integrate also a paragraph on the outdoor thermal comfort? E.g., by computing the UTCI or the PET. It would be also nice to evaluate the MRT, which is also an output from the tool.

What is the limit on the height of buildings, upon that the cooling effect is negligible? Is there any relationship between the wind direction and the main orientation of buildings?

The test points indicated at p. 9 are the one from Figure 5?

I propose to add a final Table in order to summarise the results.

Author Response

To reviewer 3#

This paper aims to understand the impact of green roofs on the outdoor thermal comfort. The paper is really interesting, leading to an important topic for the sustainable urban development. The study is quite comprehensive, combining both the measurements with the simulations with ENVImet.

Response: Many thanks go to your comments and we have revised this paper according to your comments.

Q1: In order to evaluate the impact of green roofs on the outdoor thermal comfort, the air temperature is computed. As you have already the model in ENVImet, can you integrate also a paragraph on the outdoor thermal comfort? E.g., by computing the UTCI or the PET. It would be also nice to evaluate the MRT, which is also an output from the tool.

Response 1: In this paper, we have not mentioned the outdoor thermal comfort, but we will study this in the future.

Q2: What is the limit on the height of buildings, upon that the cooling effect is negligible? Is there any relationship between the wind direction and the main orientation of buildings?

Response 2: We have discussed the impacts of wind direction and building orientation on building height limits. Please be kind to check Line 387-389, Line 413-422. Line 475-478.

Q3: The test points indicated at p. 9 are the one from Figure 5?

Response 3: Yes, it is. We have revised the Figure 6, as Point A -Point G.

Q4: I propose to add a final Table in order to summarise the results.

Response 4: Yes, we have added the Table 3.

Round 2

Reviewer 1 Report

Better and suitable for publication.

It is still not clear on that the effects of green roofs on pedestrian comfort is limited and max AIR temperature change is small.

Reviewer 2 Report

I reviewed all the review comments. Authors have revised the article considerably with major rewriting at many places and added clarity where needed.